# Psychological morbidity among children with transfusion dependent β-thalassaemia and their parents in Sri Lanka

**Sachith Mettananda**[1,2]*, **Ravindu Peiris**[1], **Hashan Pathiraja**[1], **Miyuru Chandradasa**[2,3], **Dayananda Bandara**[4], **Udaya de Silva**[5], **Chamila Mettananda**[6], **Anuja Premawardhena**[2,7]

**1** Department of Paediatrics, Faculty of Medicine, University of Kelaniya, Ragama, Sri Lanka, **2** Colombo North Teaching Hospital, Ragama, Sri Lanka, **3** Department of Psychiatry, Faculty of Medicine, University of Kelaniya, Ragama, Sri Lanka, **4** Kurunegala Teaching Hospital, Kurunegala, Sri Lanka, **5** Anuradhapura Teaching Hospital, Anuradhapura, Sri Lanka, **6** Department of Pharmacology, Faculty of Medicine, University of Kelaniya, Kelaniya, Sri Lanka, **7** Department of Medicine, Faculty of Medicine, University of Kelaniya, Kelaniya, Sri Lanka

* sachith.mettananda@kln.ac.lk

**Data Availability Statement:** All relevant data are within the paper.

## Abstract

### Background

Thalassaemia is a chronic disease which requires lifelong treatment in a majority. Despite recent advances in the medical care, minimal attempts are made to improve psychological health in these patients. In this study, we aim to describe the psychological morbidity in patients with transfusion dependent β-thalassaemia and their mothers in Sri Lanka.

### Methods

This case control study was conducted in the three largest thalassaemia centres of Sri Lanka. All patients with transfusion dependent β-thalassaemia aged 4–18 years were recruited as cases whilst a randomly selected group of children without chronic diseases were recruited as controls. Psychological morbidity of children was assessed using the Strengths and Difficulties Questionnaire and depressive symptoms of mothers was assessed by the Centre for Epidemiological Studies Depression Scale.

### Results

288 transfusion dependent β-thalassaemia patients and equal number of controls were recruited. Abnormal emotional, conduct, hyperactivity and peer relationship symptom scores were reported by 18%, 17%, 9% and 14% of patients with thalassaemia respectively. Prevalences of abnormal psychological symptom scores in all domains were significantly higher among patients compared to controls. Abnormal conduct symptoms were significantly more prevalent among patients with HbE β-thalassaemia and those with suboptimal pretransfusion haemoglobin levels, lower transfusion volumes, hypothyroidism and undernutrition. Short stature was associated with abnormal emotional and hyperactivity scores. Depressive symptoms were significantly higher among mothers of patients with

**Funding:** The author(s) received no specific funding for this work.

**Competing interests:** The authors have declared that no competing interests exist.

thalassaemia. Higher depressive symptom scores in mothers were significantly associated with abnormal emotional, conduct and peer relationship symptom scores in children.

## Conclusions

A higher proportion of patients with transfusion dependent β-thalassaemia had abnormal psychological symptom scores. Abnormal conduct symptoms were more prevalent among patients with HbE β-thalassaemia, those who were inadequately transfused and having hypothyroidism and undernutrition. Mothers of the children with transfusion dependent β-thalassaemia had significantly higher depressive symptoms which were significantly associated with psychological symptoms among children.

## Introduction

β-thalassaemia is one of the most common monogenic diseases in the world[1, 2]. Approximately 70,000 children are born with various forms of thalassaemia each year[3]. Except for a minority who are cured by haematopoietic stem cell transplantation, all patients with severe forms of β-thalassaemia require regular blood transfusions and iron chelator medication for life[4, 5]. They have shorter life expectancies and poor quality of life[6].

The prevalence of β-thalassaemia is highest in the tropical regions extending from Mediterranean to south and southeast Asia[7]. Sri Lanka is a south Asian country located within this tropical thalassaemia belt. β-thalassaemia gene frequency in Sri Lanka is reported as 2.8% whilst 1800 patients with transfusion dependent β-thalassaemia (TDBT) are being treated in twenty-six thalassaemia centres across the country[8]. However, over sixty percent of these patients are managed in the three largest thalassaemia referral centres in Kurunegala, Anuradhapura and Ragama[9].

Patients with TDBT are invariably at risk of developing increased stress and psychological issues. Although, the current emphasis is mostly on improving medical management of these patients and devising a cure, minimal efforts are made to uplift their psychological health[10–14]. In fact, the burden of phycological morbidity among patients with thalassaemia have not been well documented. Early detection of mental health issues using psychometric instruments is essential in the current Sri Lankan context, as the child and adolescent psychiatry expertise are limited [15, 16]. In this study we aim to describe the psychological morbidity in a cohort of patients with TDBT and their mothers in Sri Lanka.

## Methods

We conducted a case control study from September 2017 to March 2018 in the three largest thalassaemia centres of Sri Lanka situated in Kurunegala, Anuradhapura and Ragama hospitals. All patients with TDBT who are aged between 4 to 18 years attending these centres were recruited as cases. The diagnosis of thalassaemia was confirmed by haemoglobin high performance liquid chromatography and 'transfusion dependency' was defined as receiving more than eight blood transfusion during preceding twelve months. This group comprised of more than half the children with TDBT aged between 4 to 18 year in Sri Lanka. A group of children without chronic diseases who were admitted to the same hospitals for acute non-life-threatening illnesses were recruited as controls. Controls were selected by simple random sampling using a random number table until same number of controls as cases were recruited. All

**Table 1. Clinical characteristics of patients with transfusion dependent β- thalassaemia.**

| Characteristic | N = 288 |
|---|---|
| | **Mean (±SD)** |
| Duration of the disease (years) | 9.2 (±3.9) |
| Average pretransfusion haemoglobin (g/dL) | 8.3 (±1.2) |
| Annual transfusion requirement (ml/kg/year) | 233 (±74) |
| Average serum ferritin (ng/mL) | 1942 (±1798) |
| Duration of hospital stay (days) | 2.20 (±0.96) |
| Distance from home to hospital (km) | 55.2 (±50.9) |
| | **Frequency (%)** |
| Age groups | |
| 4–7 years | 88 (30.3%) |
| 8–12 years | 118 (41.0%) |
| 13–18 years | 82 (28.5%) |
| Sex | |
| Male | 138 (47.9%) |
| Female | 150 (52.1%) |
| Sub-type of thalassaemia | |
| β-thalassaemia major | 238 (82.6%) |
| HbE β-thalassaemia | 47 (16.4%) |
| Other | 3 (1.0%) |
| Thalassaemia treatment centre | |
| Kurunegala | 170 (59.0%) |
| Anuradhapura | 77 (26.7%) |
| Ragama | 41 (14.2%) |
| Frequency of blood transfusions | |
| > 4 weekly | 33 (11.5%) |
| 4 weekly | 219 (76.0%) |
| 3 weekly | 35 (12.2%) |
| < 3 weekly | 1 (0.3%) |
| Average pretransfusion haemoglobin | |
| < 7.0 g/dl | 39 (13.5%) |
| 7.0–8.9 g/dl | 139 (48.3%) |
| 9.0–10.5 g/dl | 101 (35.1%) |
| > 10.5 g/dl | 9 (3.1%) |
| Annual transfusion requirement[1] | |
| < 200 ml/kg/year | 121 (44.0%) |
| 201–250 ml/kg/year | 43 (15.6%) |
| 251–300 ml/kg/year | 59 (21.5%) |
| > 300 ml/kg/year | 52 (18.9%) |
| Spleen status | |
| No splenomegaly | 189 (65.6%) |
| Splenomegaly of 1–3 cm | 80 (27.8%) |
| Splenomegaly of > = 4 cm | 12 (4.2%) |
| Splenectomised | 7 (2.4%) |
| Liver status | |
| No hepatomegaly | 199 (69.1%) |
| Hepatomegaly of 1–2 cm | 73 (25.3%) |
| Hepatomegaly > = 3cm | 16 (5.6%) |

(*Continued*)

**Table 1.** (Continued)

| Characteristic | N = 288 |
|---|---|
| Serum Ferritin[2] | |
| < 1000 ng/mL | 91 (32.7%) |
| 1001–2500 ng/mL | 128 (46.0%) |
| 2501–5000 ng/mL | 38 (13.7%) |
| > 5000 ng/mL | 21 (7.6%) |
| Iron chelator medication | |
| No chelation | 1 (0.3%) |
| Deferasirox | 178 (61.8%) |
| Deferoxamine | 30 (10.4% |
| Deferiprone | 3 (1.0%) |
| Deferasirox + Deferoxamine | 76 (26.4%) |
| Complications | |
| Thalassaemia facies | 94 (32.6%) |
| Skin pigmentation | 58 (20.1%) |
| Short stature[3] | 115 (44.1%) |
| Undernutrition[4] | 88 (34.0%) |
| Type 1 diabetes | 3 (1.0%) |
| Hypothyroidism | 11 (3.8%) |
| Cardiomyopathy | 2 (0.7%) |
| Elevated transaminases | 54 (18.8%) |
| Cirrhosis | 0 |
| Allergic reaction to transfusion | 60 (20.8%) |
| Hepatitis C Infection | 64 (22.2%) |
| Abdominal scars | 66 (22.9%) |
| Hearing impairment | 3 (1.0%) |
| Visual impairment | 20 (6.9%) |

Data missing from:

[1]13 patients;

[2]10 patients;

[3]27 patients; and

[4]29 patients

participants were recruited after obtaining informed written consent from guardians and assent from children over 12 years.

Data were collected using several study instruments. Firstly, an interviewer-administered questionnaire was completed by a trained data collector by interviewing patients and their parents to gather data on socio-demographics. Secondly, a data collection form was completed from cases by perusal of clinical records and physical examinations by trained doctors to gather information on pre-transfusion haemoglobin levels, volume of blood transfusions, anthropometric measurements, liver and spleen sizes and disease complications. Psychological morbidity of children was then assessed using the Strengths and difficulties questionnaire which measures 25 attributes, along five scales; emotional, conduct, hyperactivity, peer relationship problems and pro-social behaviours. This is a parent-administered questionnaire which has been previously translated, used and validated in Sinhalese[17–19]. Depressive symptoms of parents were assessed by the 20-item 'Centre for epidemiological studies depression scale (CESD)' which has also been previously translated and validated for Sinhalese[20].

Data were analysed using IBM SPSS statistics 25.0 for windows. Categorical data were expressed as frequencies (percentages) and continuous data were expressed as median (inter-quartile range) and mean (standard deviations). Mann-Whitney U test was used to compare medians and $\chi^2$-test and binary logistic regression method were used to identify significant associations between categorical variables. Statistical significance was defined as $p < 0.05$. Ethical approval was obtained from the Ethics Review Committee of University of Kelaniya, Sri Lanka.

## Results

A total of 288 patients and an equal number of controls were recruited into the study. Age and sex distributions of patients and controls were similar. Mean ages were 10.3±3.8 and 9.7±3.7 years respectively for patients and controls (p = 0.054). 138 (47.9%) patients and 134 (46.5%) controls were males ($\chi^2 = 0.73$, p = 0.80). Of the patients, 238 (82.6%) had homozygous β-thalassaemia major, 47 (16.4%) had haemoglobin (Hb) E β-thalassaemia and three (1.0%) patients had other forms of transfusion dependent β-thalassaemia. Clinical characteristics of the patient population are shown in Table 1.

### Psychological symptom scores among patients with TDBT

Behavioural and emotional symptoms as reported by the parents among patients with TDBT were assessed using previously validated strengths and difficulties questionnaire. Abnormal emotional, conduct and hyperactivity symptom scores were reported by 18%, 17% and 9% of children with TDBT respectively (Table 2). 14% had abnormal peer relationship scores whilst 2% had abnormal prosocial scores. Prevalences of abnormal symptom scores were significantly higher among patients with TDBT compared to controls in all domains (Table 3). Patients with HbE β-thalassaemia reported significantly higher prevalence of abnormal conduct symptom scores compared to the subset with β-thalassaemia major. There were no significant differences in psychological symptoms scores in other domains among patients with β-thalassaemia major and HbE β-thalassaemia.

### Clinical and socio-demographic factors associated with abnormal psychological symptoms scores

Next, we examined for clinical and socio-demographic factors associated with abnormal psychological symptom scores among patients with TDBT using logistic regression. A significantly higher proportion of children with suboptimal pretransfusion haemoglobin levels had abnormal conduct and peer relationship scores (Table 4). Similarly, higher proportion of children who received lower annual transfusion volumes (less than 250 ml/kg/year) reported abnormal conduct, hyperactivity and total symptom scores. Presence of hypothyroidism and undernutrition were significantly associated with abnormal conduct scores, thalassaemic facies and undernutrition was associated with abnormal total symptom scores and short stature was associated with abnormal emotional and hyperactivity scores. The analysis of socio-demographic factors revealed that lower family income and living far away from the treatment centre has significant associations with abnormal hyperactivity symptom score and peer relationship score respectively (Table 5).

### Depressive symptoms among mothers of children with TDBT

Next, we analysed the depressive symptoms of mothers of patients with TDBT using Centre for Epidemiological Studies Depression Scale (CESD). Depressive symptom scores were

**Table 2. Distribution of psychological health symptom scores of patients with TDBT.**

| Psychological health parameter | Normal N (%) | Borderline N (%) | Abnormal N (%) |
|---|---|---|---|
| Emotional symptom score | 191 (66.3%) | 45 (15.6%) | 52 (18.1%) |
| Conduct symptom score | 173 (60.1%) | 65 (22.6%) | 50 (17.4%) |
| Hyperactivity symptom score | 231 (80.2%) | 31 (10.8%) | 26 (9.0%) |
| Peer relationship score | 207 (71.9%) | 41 (14.2%) | 40 (13.9%) |
| Prosocial score | 263 (91.3%) | 19 (6.6%) | 6 (2.1%) |
| Total score | 204 (70.8%) | 40 (13.9%) | 44 (15.3%) |

significantly higher among mothers with patients with TDBT compared to controls (p<0.001; Mann-Whitney test) (Fig 1). A large proportion (121/285; 42.5%) of mothers of children with TDBT had abnormally high depressive symptom scores (CESD score > = 21; for the Sinhalese version which has a sensitivity of 73% and specificity of 96%)[20].

Then we examined the association between clinical characteristics of patients with TDBT and depressive symptom scores in mothers. This analysis revealed that receipt of low annual transfusion volumes (less than 250mL/kg/year) and the presence of hypothyroidism in children with TDBT were significantly associated with higher depressive symptom scores in mothers (Table 6). The presence of undernutrition among patients was significantly associated with lower maternal depression symptom scores in this analysis.

## Association between maternal depressive symptoms and abnormal psychological symptom scores among patients with TDBT

Finally, we examined the relationship between maternal depression and psychological symptoms among patients with TDBT. This revealed that higher depressive symptom scores in mothers were significantly associated with abnormal psychological health scores in emotional symptoms, conduct symptoms, peer relationships and total symptoms scores in children (Table 7).

**Table 3. Prevalence of abnormal psychological health symptoms scores among patients with TDBT and controls.**

| Psychological health parameter | Patients with TDBT (n = 288) | Healthy controls (n = 288) | OR (95%CI) | p |
|---|---|---|---|---|
| Abnormal emotional symptom score | 52 (18.1%) | 2 (0.7%) | 31.5 (7.5–130.7) | <0.001 |
| Abnormal conduct symptom score | 50 (17.4%) | 5 (1.7%) | 11.8 (4.6–30.2) | <0.001 |
| Abnormal hyperactivity symptom score | 26 (9.0%) | 9 (3.1%) | 3.07 (1.41–6.68) | <0.01 |
| Abnormal peer relationship score | 40 (13.9%) | 10 (3.5%) | 4.48 (2.19–9.15) | <0.001 |
| Abnormal prosocial score | 6 (2.1%) | 0 | | <0.05* |
| Abnormal total score | 44 (15.3%) | 3 (1.0%) | 17.1 (5.2–55.8) | <0.001 |
| | β-thalassaemia major (n = 238) | haemoglobin E β-thalassaemia (n = 47) | OR (95%CI) | p |
| Abnormal emotional symptom score | 45 (18.9%) | 6 (12.8%) | 0.62 (0.25–1.56) | 0.31 |
| Abnormal conduct symptom score | 35 (14.7%) | 13 (27.7%) | 2.21 (1.06–4.61) | <0.05 |
| Abnormal hyperactivity symptom score | 21 (8.8%) | 4 (8.5%) | 0.96 (0.31–2.94) | 0.94 |
| Abnormal peer relationship score | 29 (12.2%) | 9 (19.1%) | 1.70 (0.74–3.89) | 0.19 |
| Abnormal prosocial score | 5 (2.1%) | 0 | | 0.59* |
| Abnormal total score | 36 (15.1%) | 7 (14.9%) | 0.98 (0.40–1.36) | 0.96 |

*Fisher's exact test

**Table 4. Associations between clinical characteristics and abnormal psychological health symptoms scores among patients with transfusion dependent β-thalassaemia[1].**

| | Abnormal emotional symptom score | Abnormal conduct symptom score | Abnormal hyperactivity symptom score | Abnormal peer relationship score | Abnormal prosocial score | Abnormal total score |
|---|---|---|---|---|---|---|
| Average pretransfusion haemoglobin | | | | | | |
| > 9.0 g/dl (n = 90) | 13 (14.4%) | 10 (11.1%) | 10 (11.1%) | 6 (6.7%) | 2 (2.2%) | 8 (8.9%) |
| < 9.0 g/dl (n = 153) | 29 (19.0%) | 30 (19.6%) | 15 (9.8%) | 26 (17.0%) | 3 (2.0%) | 26 (17.0%) |
| Adjusted odds ratios (95%CI) | 1.54 (0.69–3.43) | 2.82 (1.17–6.80) | 1.14 (0.43–3.03) | 2.76 (1.01–7.54) | 1.32 (0.18–9.63) | 2.49 (0.96–6.46) |
| *p-value (adjusted)* | *0.29* | *<0.05* | *0.78* | *<0.05* | *0.78* | *0.05* |
| Annual transfusion volumes | | | | | | |
| > 250 ml/kg/year (n = 101) | 14 (13.9%) | 11 (10.9%) | 3 (3.0%) | 11 (10.9%) | 2 (2.0%) | 10 (9.9%) |
| < 250 ml/kg/year (n = 142) | 28 (19.7%) | 29 (20.4%) | 22 (15.5%) | 21 (14.8%) | 3 (2.1%) | 24 (16.9%) |
| Adjusted odds ratios (95%CI) | 2.16 (0.94–4.93) | 2.98 (1.24–7.13) | 8.89 (2.27–34.7) | 1.69 (0.69–4.12) | 1.51 (0.19–11.7) | 3.05 (1.16–7.98) |
| *p-value (adjusted)* | *0.06* | *<0.05* | *<0.01* | *0.24* | *0.68* | *<0.05* |
| Liver status | | | | | | |
| No hepatomegaly (n = 169) | 26 (15.4%) | 25 (14.8%) | 16 (9.5%) | 17 (10.1%) | 4 (2.4%) | 22 (13.0%) |
| Hepatomegaly (n = 74) | 16 (21.6%) | 15 (20.3%) | 9 (12.2%) | 15 (20.3%) | 1 (1.4%) | 12 (16.2%) |
| Adjusted odds ratios (95%CI) | 1.31 (0.48–3.57) | 1.71 (0.61–4.77) | 1.09 (3.16–3.82) | 2.17 (0.72–6.52) | 0.72 (0.04–12.5) | 1.03 (0.33–3.20) |
| *p-value (adjusted)* | *0.59* | *0.30* | *0.88* | *0.16* | *0.82* | *0.95* |
| Spleen status | | | | | | |
| No splenomegaly (n = 167) | 26 (15.6%) | 25 (15.0%) | 16 (9.6%) | 17 (10.2%) | 4 (2.4%) | 21 (12.6%) |
| Splenomegaly (n = 76) | 16 (21.1%) | 15 (19.7%) | 9 (11.8%) | 15 (19.7%) | 1 (1.3%) | 13 (17.1%) |
| Adjusted odds ratios (95%CI) | 1.09 (0.39–3.00) | 1.41 (0.50–3.98) | 1.02 (0.30–3.49) | 1.10 (0.37–3.28) | 1.03 (0.06–17.7) | 1.16 (0.38–3.58) |
| *p-value (adjusted)* | *0.86* | *0.50* | *0.97* | *0.86* | *0.98* | *0.78* |
| Serum ferritin | | | | | | |
| > 1000 ng/mL (n = 163) | 23 (14.1%) | 23 (14.1%) | 17 (10.4%) | 19 (11.7%) | 4 (2.5%) | 18 (11.0%) |
| < 1000 ng/mL (n = 80) | 19 (23.8%) | 17 (21.3%) | 8 (10.0%) | 13 (16.3%) | 1 (1.3%) | 16 (20.0%) |
| Adjusted odds ratios (95%CI) | 1.70 (0.83–3.50) | 1.43 (0.68–3.01) | 0.81 (0.31–2.12) | 1.36 (0.61–3.03) | 0.52 (0.05–5.29) | 1.84 (0.83–4.09) |
| *p-value (adjusted)* | *0.14* | *0.34* | *0.67* | *0.44* | *0.58* | *0.13* |
| Thalassaemic facies | | | | | | |
| No (n = 159) | 22 (13.8%) | 26 (16.4%) | 13 (8.2%) | 18 (11.3%) | 5 (3.1%) | 14 (8.8%) |
| Yes (n = 84) | 20 (23.8%) | 14 (16.7%) | 12 (14.3%) | 14 (16.7%) | 0 (0.0%) | 20 (23.8%) |
| Adjusted odds ratios (95%CI) | 1.80 (0.85–3.82) | 0.87 (0.39–1.94) | 1.96 (0.76–5.04) | 1.39 (0.60–3.22) | 0.00 | 3.42 (1.48–7.90) |
| *p-value (adjusted)* | *0.12* | *0.75* | *0.15* | *0.44* | *0.99* | *<0.01* |
| Skin pigmentation | | | | | | |
| No (n = 201) | 35 (17.4%) | 33 (16.4%) | 21 (10.4%) | 27 (13.4%) | 5 (2.5%) | 28 (13.9%) |
| Yes (n = 42) | 7 (16.7%) | 7 (16.7%) | 4 (9.5%) | 5 (11.9%) | 0 (0.0%) | 6 (14.3%) |
| Adjusted odds ratios (95%CI) | 0.83 (0.31–2.21) | 0.82 (0.30–2.24) | 0.65 (0.18–2.27) | 0.84 (0.28–2.53) | 0.00 | 0.78 (0.26–2.32) |
| *p-value (adjusted)* | *0.71* | *0.70* | *0.50* | *0.76* | *0.99* | *0.66* |

(*Continued*)

**Table 4.** (Continued)

| | Abnormal emotional symptom score | Abnormal conduct symptom score | Abnormal hyperactivity symptom score | Abnormal peer relationship score | Abnormal prosocial score | Abnormal total score |
|---|---|---|---|---|---|---|
| Diabetes | | | | | | |
| No (n = 241) | 41 (17.0%) | 39 (16.2%) | 25 (10.4%) | 31 (12.9%) | 5 (2.1%) | 33 (13.7%) |
| Yes (n = 2) | 1 (50.0%) | 1 (50.0%) | 0 (0.0%) | 1 (50.0%) | 0 (0.0%) | 1 (50.0%) |
| Adjusted odds ratios (95%CI) | 1.89 (0.09–37.4) | 0.90 (0.04–19.0) | 0.0 | 8.62 (0.37–196) | 0.00 | 2.43 (0.11–50.8) |
| *p-value (adjusted)* | *0.67* | *0.94* | *0.99* | *0.17* | *0.99* | *0.56* |
| Hypothyroidism | | | | | | |
| No (n = 232) | 39 (16.8%) | 35 (15.1%) | 24 (10.3%) | 30 (12.9%) | 5 (2.2%) | 31 (13.4%) |
| Yes (n = 11) | 3 (27.3%) | 5 (45.5%) | 1 (9.1%) | 2 (18.2%) | 0 (0.0%) | 3 (27.3%) |
| Adjusted odds ratios (95%CI) | 1.95 (0.42–8.94) | 5.94 (1.46–24.0) | 0.86 (0.09–8.35) | 1.04 (0.16–6.83) | 0.00 | 2.71 (0.56–12.9) |
| *p-value (adjusted)* | *0.38* | *<0.05* | *0.90* | *0.96* | *0.99* | *0.21* |
| Short stature | | | | | | |
| No (n = 138) | 15 (10.9%) | 17 (12.3%) | 9 (6.5%) | 20 (14.5%) | 3 (2.2%) | 14 (10.1%) |
| Yes (n = 105) | 27 (25.7%) | 23 (21.9%) | 16 (15.2%) | 12 (11.4%) | 2 (1.9%) | 20 (19.0%) |
| Adjusted odds ratios (95%CI) | 2.56 (1.21–5.41) | 1.89 (0.89–4.04) | 2.83 (1.10–7.35) | 0.68 (0.29–1.58) | 1.14 (0.16–7.82) | 1.81 (0.78–4.16) |
| *p-value (adjusted)* | *<0.05* | *0.09* | *<0.05* | *0.37* | *0.89* | *0.16* |
| Undernutrition | | | | | | |
| No (n = 158) | 22 (13.9%) | 22 (13.9%) | 16 (10.1%) | 24 (15.2%) | 2 (1.3%) | 18 (11.4%) |
| Yes (n = 85) | 20 (23.5%) | 18 (21.2%) | 9 (10.6%) | 8 (9.4%) | 3 (3.5%) | 16 (18.8%) |
| Adjusted odds ratios (95%CI) | 2.08 (0.98–4.38) | 2.24 (1.03–4.83) | 1.32 (0.50–3.44) | 0.67 (0.27–1.66) | 1.14 (0.16–7.82) | 2.37 (1.02–5.47) |
| *p-value (adjusted)* | *0.05* | *<0.05* | *0.56* | *0.38* | *0.89* | *<0.05* |
| Hepatitis C infection | | | | | | |
| No (n = 184) | 31 (16.8%) | 31 (16.8%) | 19 (10.3%) | 22 (12.0%) | 4 (2.2%) | 25 (13.6%) |
| Yes (n = 59) | 11 (18.6%) | 9 (15.3%) | 6 (10.2%) | 10 (16.9%) | 1 (1.7%) | 9 (15.3%) |
| Adjusted odds ratios (95%CI) | 1.42 (0.57–3.54) | 1.31 (0.50–3.42) | 1.64 (0.50–5.39) | 1.35 (0.50–3.63) | 1.51 (0.13–16.3) | 1.43 (0.51–4.00) |
| *p-value (adjusted)* | *0.44* | *0.58* | *0.41* | *0.54* | *0.73* | *0.48* |

[1]Data of 243 patients with completed data were analysed using binary logistic regression.

## Discussion

In this paper, we examined the psychological health among children with TDBT and their mothers. We studied almost half of the TDBT patient population in Sri Lanka in the specified age group attending the three main thalassaemia centres of Sri Lanka which are situated in three different provinces of the country[9]. We found that children with TDBT in Sri Lanka have significantly higher prevalence of psychological symptoms in emotional, conduct and hyperactivity domains and abnormal peer relationships and social skills. This confirms findings of a previous study done among 60 Iranian children with thalassaemia major who reported higher rates of behavioural problems in emotional, conduct, hyperactivity, peer relationship, social and all states compared to healthy children[21]. Similar findings were reported in few other small-scale studies[22, 23]. However, ours is the largest of such studies. Similarly, most previous studies were done in eras where the care of patients with β-thalassaemia was not well developed and in times patients were receiving iron chelator medications through lengthy subcutaneous infusions. Our results show that only minimal improvements in psychological

**Table 5. Associations between socio-demographic characteristics and abnormal psychological health symptoms scores among patients with transfusion dependent β-thalassaemia[1].**

| | Abnormal emotional symptom score | Abnormal conduct symptom score | Abnormal hyperactivity symptom score | Abnormal peer relationship score | Abnormal prosocial score | Abnormal total score |
|---|---|---|---|---|---|---|
| Education level of the mother | | | | | | |
| Below O/L (n = 180) | 34 (18.9%) | 31 (17.2%) | 15 (8.3%) | 29 (16.1%) | 4 (2.2%) | 30 (16.7%) |
| A/L or higher (n = 80) | 12 (15.0%) | 11 (13.8%) | 8 (10.0%) | 6 (7.5%) | 2 (2.5%) | 6 (7.5%) |
| Adjusted odds ratios (95%CI) | 0.87 (0.37–2.04) | 1.15 (0.47–2.82) | 1.13 (0.35–3.60) | 1.83 (0.63–5.31) | 1.40 (0.16–12.0) | 1.81 (0.62–5.28) |
| p-value (adjusted) | 0.74 | 0.75 | 0.82 | 0.26 | 0.75 | 0.27 |
| Education level of the father | | | | | | |
| Below O/L (n = 195) | 39 (20.0%) | 34 (17.4%) | 15 (7.7%) | 30 (15.4%) | 4 (2.1%) | 31 (15.9%) |
| A/L or higher (n = 65) | 7 (10.8%) | 8 (12.3%) | 8 (12.3%) | 5 (7.7%) | 2 (3.1%) | 5 (7.7%) |
| Adjusted odds ratios (95%CI) | 2.51 (0.84–7.46) | 1.84 (0.64–5.27) | 0.67 (0.20–2.27) | 1.10 (0.32–3.69) | 0.71 (0.07–6.78) | 1.72 (0.50–5.89) |
| p-value (adjusted) | 0.09 | 0.25 | 0.53 | 0.87 | 0.77 | 0.38 |
| Mother's occupation | | | | | | |
| Housewife/ Unemployed (n = 221) | 38 (17.2%) | 35 (15.8%) | 17 (7.7%) | 32 (14.5%) | 5 (2.3%) | 30 (13.6%) |
| Employed (n = 39) | 8 (20.5%) | 7 (17.9%) | 6 (15.4%) | 3 (7.7%) | 1 (2.6%) | 6 (15.4%) |
| Adjusted odds ratios (95%CI) | 1.95 (0.73–5.16) | 1.38 (0.51–3.72) | 2.90 (0.87–9.64) | 0.67 (0.18–2.51) | 0.72 (0.06–7.95) | 2.03 (0.69–5.97) |
| p-value (adjusted) | 0.17 | 0.52 | 0.08 | 0.55 | 0.79 | 0.19 |
| Father's occupation | | | | | | |
| Unemployed/ Unskilled (n = 180) | 32 (17.8%) | 27 (15.0%) | 13 (7.2%) | 28 (15.6%) | 4 (2.2%) | 26 (14.4%) |
| Skilled/ Professional (n = 80) | 14 (17.5%) | 15 (18.8%) | 10 (12.5%) | 7 (8.8%) | 2 (2.5%) | 10 (12.5%) |
| Adjusted odds ratios (95%CI) | 0.55 (0.23–1.33) | 0.56 (0.24–1.34) | 0.43 (0.14–1.32) | 1.63 (0.57–4.66) | 2.43 (0.31–18.5) | 0.53 (0.20–1.39) |
| p-value (adjusted) | 0.18 | 0.19 | 0.14 | 0.35 | 0.39 | 0.19 |
| Monthly family income (LKR) | | | | | | |
| < 25,000 (n = 175) | 35 (20.0%) | 28 (16.0%) | 17 (9.7%) | 27 (15.4%) | 2 (1.1%) | 29 (16.6%) |
| > 25,000 (n = 85) | 11 (12.9%) | 14 (16.5%) | 6 (7.1%) | 8 (9.4%) | 4 (4.7%) | 7 (8.2%) |
| Adjusted odds ratios (95%CI) | 1.95 (0.77–4.92) | 1.02 (0.43–2.46) | 4.03 (1.05–15.4) | 1.63 (0.57–4.66) | 0.16 (0.02–1.12) | 2.32 (0.80–6.71) |
| p-value (adjusted) | 0.15 | 0.94 | <0.05 | 0.35 | 0.66 | 0.11 |
| Average duration of hospital stay | | | | | | |
| 1 day (n = 46) | 6 (13.0%) | 6 (13.0%) | 7 (15.2%) | 7 (15.2%) | 2 (4.3%) | 3 (6.5%) |
| >1 day (n = 214) | 40 (18.7%) | 36 (16.8%) | 16 (7.5%) | 28 (13.1%) | 4 (1.9%) | 33 (15.4%) |
| Adjusted odds ratios (95%CI) | 1.32 (0.49–3.54) | 1.37 (0.51–3.65) | 0.51 (0.17–1.49) | 0.51 (0.19–1.37) | 0.43 (0.06–2.79) | 2.18 (0.80–6.71) |
| p-value (adjusted) | 0.57 | 0.52 | 0.22 | 0.18 | 0.38 | 0.11 |
| Distance from home to hospital | | | | | | |
| < 50 km (n = 170) | 24 (14.1%) | 25 (14.7%) | 16 (9.4%) | 17 (10.0%) | 4 (2.4%) | 19 (11.2%) |
| >50 km (n = 90) | 22 (24.4%) | 17 (18.9%) | 7 (7.8%) | 18 (20.0%) | 2 (2.2%) | 17 (18.9%) |

*(Continued)*

**Table 5.** (Continued)

| | Abnormal emotional symptom score | Abnormal conduct symptom score | Abnormal hyperactivity symptom score | Abnormal peer relationship score | Abnormal prosocial score | Abnormal total score |
|---|---|---|---|---|---|---|
| Adjusted odds ratios (95%CI) | 1.82 (0.93–3.54) | 1.25 (0.63–2.51) | 0.81 (0.30–2.19) | 2.32 (1.09–4.94) | 1.22 (0.19–7.87) | 1.57 (0.75–3.27) |
| p-value (adjusted) | 0.07 | 0.51 | 0.69 | <0.05 | 0.82 | 0.22 |

[1]Data of 260 patients with completed data were analysed using binary logistic regression.

aspects of patients with TDBT are made despite remarkable improvement in medical management with safe blood products and wide-spread availability of oral iron chelators. We previously reported lower health related quality of life scores among children with TDBT[6]. Results presented here well collaborate with the findings of that study and demonstrate that the effects are not limited to quality of life but extends to psychological symptoms as well.

We found that patients with HbE β-thalassaemia have abnormal conduct symptom scores compared to patients with β-thalassaemia major. We previously showed that these patients receive suboptimal transfusions and maintain lower pre-transfusion haemoglobin levels and also have lower quality of life[6, 24]. Therefore, the higher prevalence of conduct disorder symptoms in HbE β-thalassaemia patients could be related to one or all of these factors.

In logistic regression, we found higher prevalences of psychological symptoms among patients who maintain lower pre-transfusion haemoglobin levels (conduct and peer relationship symptoms) and those who receive lower transfusion volumes (conduct and hyperactivity symptoms). This emphasises the need for adequate blood transfusions among patients with

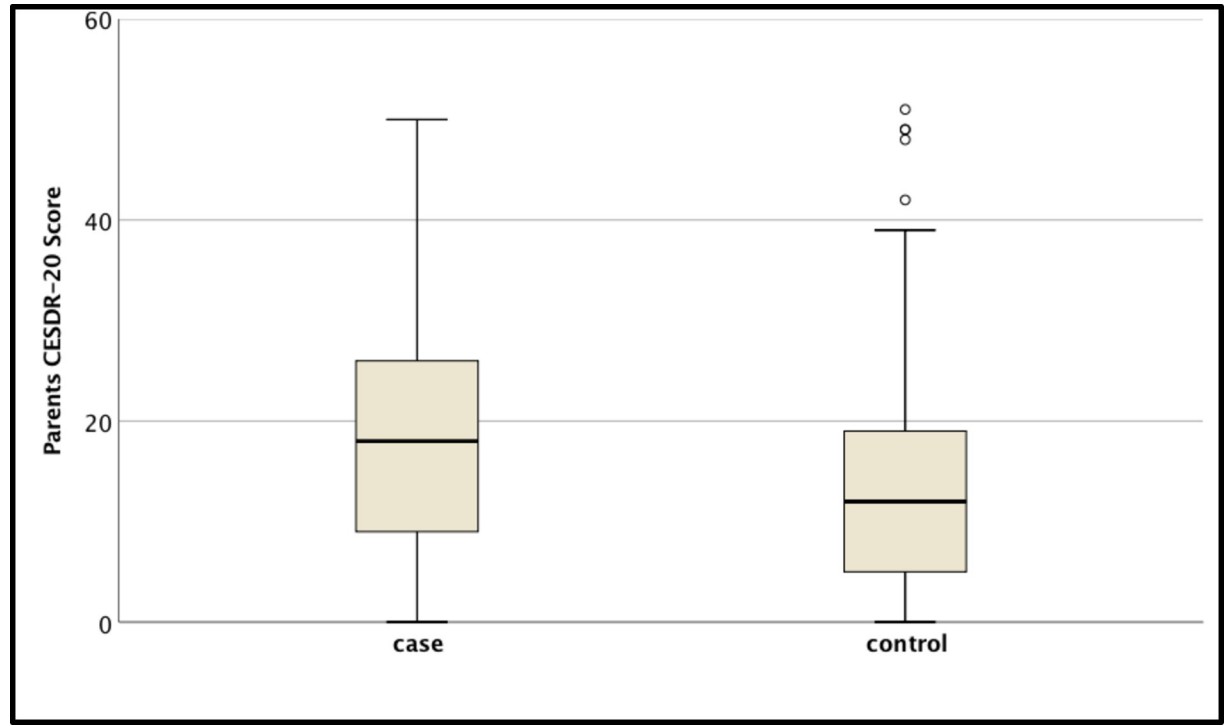

**Fig 1. Comparison of the distribution of CESD scale scores among mothers of patients with TDBT and controls.** Each box plot shows interquartile range, middle horizontal bars demonstrate respective median and error bars show range; outliers are marked in circles.

**Table 6. Associations between maternal depressive symptoms scores and clinical characteristics among patients with transfusion dependent β-thalassaemia.**

| Clinical characteristics | Higher maternal depression symptom score (n = 121) | Lower maternal depression symptom score (n = 164) | OR (95%CI) | p |
|---|---|---|---|---|
| Average pretransfusion haemoglobin <9.0 g/dl | 80 (66.1%) | 96 (58.5%) | 1.38 (0.84–2.25) | 0.19 |
| Annual transfusion volume <250 ml/kg/year[1] | 78 (66.7%) | 84 (54.2%) | 1.69 (1.02–2.78) | <0.05 |
| Hepatomegaly | 34 (28.1%) | 55 (33.5%) | 0.77 (0.46–1.29) | 0.32 |
| Splenomegaly | 38 (31.4%) | 54 (32.9%) | 0.93 (0.56–1.54) | 0.78 |
| Serum ferritin >1000 ng/mL[2] | 76 (66.7%) | 109 (67.3%) | 0.97 (0.58–1.61) | 0.91 |
| Thalassaemic facies | 42 (34.7%) | 52 (31.7%) | 1.14 (0.69–1.88) | 0.59 |
| Skin pigmentation | 27 (22.3%) | 31 (18.9%) | 1.23 (0.69–2.20) | 0.48 |
| Diabetes | 0 | 3 (1.8%) | | 0.26* |
| Hypothyroidism | 8 (6.6%) | 3 (1.8%) | 3.79 (0.98–14.6) | <0.05 |
| Short stature[3] | 47 (43.1%) | 67 (45.0%) | 0.92 (0.56–1.52) | 0.76 |
| Undernutrition[4] | 29 (27.1%) | 58 (38.9%) | 0.58 (0.34–1.00) | <0.05 |
| Hepatitis C infection | 26 (21.5%) | 38 (23.2%) | 0.90 (0.51–1.59) | 0.73 |

Data missing from:

[1] 13 patients;

[2] 9 patients;

[3] 27 patients; and

[4] 29 patients.

*Fisher's exact test

**Table 7. Association between higher maternal depression symptom scores and abnormal psychological health symptom scores among patients with TDBT[1].**

| Psychological health parameter | Lower maternal depression symptom score (n = 164) | Higher maternal depression symptom score (n = 121) | OR (95%CI) | p |
|---|---|---|---|---|
| Abnormal emotional symptom score | 13 (7.9%) | 39 (32.2%) | 5.52 (2.79–10.9) | <0.001 |
| Abnormal conduct symptom score | 21 (12.8%) | 29 (24.0%) | 2.14 (1.15–3.98) | <0.05 |
| Abnormal hyperactivity symptom score | 14 (8.5%) | 12 (9.9%) | 1.18 (0.52–2.65) | 0.68 |
| Abnormal peer relationship score | 13 (7.9%) | 27 (22.3%) | 3.33 (1.64–6.78) | <0.01 |
| Abnormal prosocial score | 5 (3.0%) | 1 (0.8%) | 0.26 (0.03–2.29) | 0.19 |
| Abnormal total score | 13 (7.9%) | 31 (25.6%) | 4.00 (1.99–8.04) | <0.001 |

[1] Data of 285 patients with completed maternal depression symptom scores were analysed

TDBT to maintain pretransfusion haemoglobin levels above 9g/dL. Interestingly body iron status measured with serum ferritin was not associated with psychological symptoms among patients with TDBT. A previous study from Brazil in apparently physically healthy children also could not find a relationship between peripheral markers of iron status and psychological symptom scores [25]. However, short stature and undernutrition showed positive associations with abnormal hyperactivity and conduct symptom scores respectively.

Another important aspect that we tested in this study is the association of depressive symptoms among mothers of TDBT patients. We found significantly higher depressive symptoms among mothers of patients with TDBT compared to controls. More importantly, our results show that high maternal depressive symptoms scores were significantly associated with abnormal psychological health scores in emotional, conduct and peer relationships domains in children. These results collaborate previous findings from different patient groups which showed significant associations between maternal depression and behavioural problems among children [26, 27]. All these findings emphasises the importance of incorporating psychological assessments, psychotherapeutic interventions and when required antidepressant treatment to the management of patients with β-thalassaemia.

One important limitation of our study is that we are unable to generalise the finding of this study to all centres caring for patients with thalassaemia in Sri Lanka. We previously showed that the care provided by larger specialised thalassaemia centres is better than the care provided by smaller centres in the country[9]. Therefore, it is likely that the psychological morbidity of children taking treatment from smaller thalassaemia centres is worse than what is described here.

In conclusion, a higher proportion of patients with TDBT had abnormal psychological symptom scores in many domains including emotional, conduct and hyperactive symptoms. Abnormal conduct disorder symptoms were more prevalent among patients with HbE β-thalassaemia, those who were inadequately transfused and having hypothyroidism and undernutrition. Mothers of the children with TDBT had significantly higher depressive symptoms which were significantly associated with psychological symptoms among their children.

## Author Contributions

**Conceptualization:** Sachith Mettananda, Miyuru Chandradasa, Dayananda Bandara, Udaya de Silva, Chamila Mettananda, Anuja Premawardhena.

**Data curation:** Sachith Mettananda, Ravindu Peiris, Hashan Pathiraja, Dayananda Bandara, Chamila Mettananda.

**Formal analysis:** Sachith Mettananda, Ravindu Peiris, Hashan Pathiraja, Chamila Mettananda, Anuja Premawardhena.

**Methodology:** Sachith Mettananda, Anuja Premawardhena.

**Project administration:** Sachith Mettananda, Ravindu Peiris, Hashan Pathiraja, Dayananda Bandara, Udaya de Silva.

**Resources:** Miyuru Chandradasa.

**Software:** Sachith Mettananda.

**Supervision:** Sachith Mettananda, Dayananda Bandara, Udaya de Silva, Anuja Premawardhena.

**Validation:** Sachith Mettananda, Miyuru Chandradasa, Chamila Mettananda.

**Visualization:** Anuja Premawardhena.

**Writing – original draft:** Sachith Mettananda, Miyuru Chandradasa, Chamila Mettananda, Anuja Premawardhena.

**Writing – review & editing:** Sachith Mettananda, Ravindu Peiris, Miyuru Chandradasa, Chamila Mettananda, Anuja Premawardhena.

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
