## [Decision Letter · Decision Letter 0]

14 Jan 2020

PONE-D-19-31863

Psychological morbidity among children with transfusion dependent beta-thalassaemia and their parents in Sri Lanka

PLOS ONE

Dear Dr Mettananda,

Thank you for submitting your manuscript to PLOS ONE. After careful consideration, we feel that it has merit but does not fully meet PLOS ONE’s publication criteria as it currently stands. Therefore, we invite you to submit a revised version of the manuscript that addresses the points raised during the review process.

The reviewers comments are very positive, and only minor discretionary revisions are required. May I suggest that the authors make these minor revisions, or justify why they cannot be made, as these will enhance the value of the paper.

We would appreciate receiving your revised manuscript by Feb 28 2020 11:59PM. To enhance the reproducibility of your results, we recommend that if applicable you deposit your laboratory protocols in protocols.io, where a protocol can be assigned its own identifier (DOI) such that it can be cited independently in the future. For instructions see: http://journals.plos.org/plosone/s/submission-guidelines#loc-laboratory-protocols

We look forward to receiving your revised manuscript.

Kind regards,

Senaka Rajapakse, MD

Academic Editor

PLOS ONE

Journal Requirements:

Additional Editor Comments (if provided):

The reviewers comments are positive, however the following two minor revisions will, I feel, enhance the value of this paper. I would recommend the authors to make these revisions and submit.

Reviewer comments:

The relationship between maternal depressive symptoms and clinical characteristics of the children (e.g.frequency of blood transfusions) would be interesting to analyze.

It would be interseting to include a short paragraph regarding the limitations of the study (if any) in the discussion section.

Reviewers' comments:

Reviewer's Responses to Questions

**Comments to the Author**

1. Is the manuscript technically sound, and do the data support the conclusions?

Reviewer #1: Yes

Reviewer #2: Yes

2. Has the statistical analysis been performed appropriately and rigorously? 

Reviewer #1: I Don't Know

Reviewer #2: Yes

3. Have the authors made all data underlying the findings in their manuscript fully available?

Reviewer #1: No

Reviewer #2: Yes

4. Is the manuscript presented in an intelligible fashion and written in standard English?

Reviewer #1: Yes

Reviewer #2: Yes

5. Review Comments to the Author

Reviewer #1: The use of a case-control study design was appropriate to meet the objectives.

The relationship between maternal depressive symptoms and clinical characteristics of the children (e.g.frequency of blood transfusions) would be interesting to analyze.

The findings are well presented and has important implications for the discipline.

It would be interseting to include a short paragraph regarding the limitations of the study (if any) in the discussion section.

Reviewer #2: Psychological morbidity among children with transfusion dependent beta-thalassaemia

and their parents in Sri Lanka

is an interesting well written article focus on important area of health,the title and abstracts were consistent with the presented problem,the background is clear and easy to understand even for those not in the field

the result are credible

the conclusion is logic and valid

6. PLOS authors have the option to publish the peer review history of their article (what does this mean?). If published, this will include your full peer review and any attached files.

Reviewer #1: No

Reviewer #2: Yes: Dr Mohamed A Yassin

---

## [Author Response · Author response to Decision Letter 0]

17 Jan 2020

Response to Reviewers’ Comments

Academic Editor Comments

The reviewer’s comments are positive, however the following two minor revisions will, I feel, enhance the value of this paper. I would recommend the authors to make these revisions and submit.

• The relationship between maternal depressive symptoms and clinical characteristics of the children (e.g. frequency of blood transfusions) would be interesting to analyze. 

• It would be interesting to include a short paragraph regarding the limitations of the study (if any) in the discussion section.

Author Response: We have now addressed above two comments raised by the reviewers in the revised manuscript. (Please see new table 6, results section and discussion)

Reviewer #1 Comments

1. The use of a case-control study design was appropriate to meet the objectives.

Author Response: No revisions were required.

2. The relationship between maternal depressive symptoms and clinical characteristics of the children (e.g.frequency of blood transfusions) would be interesting to analyze.

Author Response: We have now included analysis of maternal depressive symptoms and clinical characteristics of children in the revised manuscript (new table 6). 

3. The findings are well presented and has important implications for the discipline.

Author Response: We thank reviewer for this positive comment. 

4. It would be interesting to include a short paragraph regarding the limitations of the study (if any) in the discussion section.

Author Response: We have now included a paragraph on limitations of the study in the discussion.

Reviewer #2 Comments

1. Psychological morbidity among children with transfusion dependent beta-thalassaemia and their parents in Sri Lanka is an interesting well written article focus on important area of health, the title and abstracts were consistent with the presented problem, the background is clear and easy to understand even for those not in the field the result are credible the conclusion is logic and valid.

Author Response: We thank reviewer for this positive comment.

---

## [Editor Report · Decision Letter 1]

23 Jan 2020

Psychological morbidity among children with transfusion dependent beta-thalassaemia and their parents in Sri Lanka

PONE-D-19-31863R1

Dear Dr. Mettananda,

We are pleased to inform you that your manuscript has been judged scientifically suitable for publication and will be formally accepted for publication once it complies with all outstanding technical requirements.

With kind regards,

Senaka Rajapakse, MD

Academic Editor

PLOS ONE

---

## [Editor Report · Acceptance letter]

29 Jan 2020

PONE-D-19-31863R1 

Psychological morbidity among children with transfusion dependent b-thalassaemia and their parents in Sri Lanka  

Dear Dr. Mettananda:

I am pleased to inform you that your manuscript has been deemed suitable for publication in PLOS ONE. Congratulations! Your manuscript is now with our production department. 

With kind regards,

on behalf of

Professor Senaka Rajapakse 

Academic Editor

PLOS ONE